# Modeling the Immune Response for Pathogenic and Nonpathogenic *Orthohantavirus* Infections in Human Lung Microvasculature Endothelial Cells

**DOI:** 10.3390/v15091806

**Published:** 2023-08-24

**Authors:** Evan P. Williams, Aadrita Nandi, Victoria Nam, Linda J. S. Allen, A. Alexandre Trindade, Michele M. Kosiewicz, Colleen B. Jonsson

**Affiliations:** 1Department of Microbiology, Immunology and Biochemistry, University of Tennessee Health Science Center, Memphis, TN 38163, USA; ewilli99@uthsc.edu; 2Department of Mathematics and Statistics, Texas Tech University, Lubbock, TX 79409, USA; aadrita.nandi@ttu.edu (A.N.); victoria.nam@ttu.edu (V.N.); linda.j.allen@ttu.edu (L.J.S.A.); alex.trindade@ttu.edu (A.A.T.); 3Department of Microbiology and Immunology, University of Louisville, Louisville, KY 40202, USA; michele.kosiewicz@louisville.edu

**Keywords:** hantaviruses, lung microvascular endothelial cells, primary cells, immune response, prospect hill, andes virus, hantaan virus

## Abstract

Hantaviruses, genus *Orthohantavirus*, family *Hantaviridae*, order *Bunyavirales*, are negative-sense, single-stranded, tri-segmented RNA viruses that persistently infect rodents, shrews, and moles. Of these, only certain virus species harbored by rodents are pathogenic to humans. Infection begins with inhalation of virus particles into the lung and trafficking to the lung microvascular endothelial cells (LMVEC). The reason why certain rodent-borne hantavirus species are pathogenic has long been hypothesized to be related to their ability to downregulate and dysregulate the immune response as well as increase vascular permeability of infected endothelial cells. We set out to study the temporal dynamics of host immune response modulation in primary human LMVECs following infection by Prospect Hill (nonpathogenic), Andes (pathogenic), and Hantaan (pathogenic) viruses. We measured the level of RNA transcripts for genes representing antiviral, proinflammatory, anti-inflammatory, and metabolic pathways from 12 to 72 h with time points every 12 h. Gene expression analysis in conjunction with mathematical modeling revealed a similar profile for all three viruses in terms of upregulated genes that partake in interferon signaling (*TLR3*, *IRF7*, *IFNB1*), host immune cell recruitment (*CXCL10*, *CXCL11*, and *CCL5*), and host immune response modulation (*IDO1*). We examined secreted protein levels of IFN-β, CXCL10, CXCL11, CCL5, and IDO in two male and two female primary HLMVEC donors at 48 and 60 h post infection. All three viruses induced similar levels of CCL5, CXCL10, and CXCL11 within a particular donor, and the levels were similar in three of the four donors. All three viruses induced different protein secretion levels for both IFN-β and IDO and secretion levels differed between donors. In conclusion, we show that there was no difference in the transcriptional profiles of key genes in primary HLMVECs following infection by pathogenic and nonpathogenic hantaviruses, with protein secretion levels being more donor-specific than virus-specific.

## 1. Introduction

The RNA viruses within the genus *Orthohantavirus*, subfamily *Mammantaviridae*, family *Hantaviridae* are negative-sense, single-stranded RNA viruses with tri-segmented genomes [1,2,3]. Hantaviruses are harbored by reservoir hosts from the order Rodentia and Eulipotyphla that persist for the life of the animal [4,5]. Infection and disease in humans occur predominantly from the inhalation or consumption of rodent excreta containing virus during peridomestic activities. While not all hantaviruses cause human disease, several species may cause one of two different syndromes, hemorrhagic fever with renal syndrome (HFRS), or hantavirus pulmonary syndrome (HPS) [6,7,8]. During the early stages of illness, HFRS and HPS cases present with high fever, hypercytokinemia caused by robust activation of the immune system, extensive vascular leakage, and thrombocytopenia [9,10,11,12,13,14]. While hemorrhage is rare, petechiae may be observed throughout the body in HFRS cases. Death is usually attributed to renal failure, which occurs in up to 40% of the cases [15,16]. In HPS, vascular leakage leads to a bilateral pulmonary edema in the lungs of patients with fatalities resulting from cardiovascular and/or respiratory failure [17,18,19].

Autopsies of HPS and HFRS cases show that hantaviruses infect the lower respiratory tract in epithelial cells, but the main target of replication are the microvascular endothelial cells (LMVEC) in the kidneys (HFRS) and the lungs (HPS) [20,21,22]. During illness, immune monitoring of patient blood sera shows a robust activation of the innate immune responses with high levels of interferons, IFN-β and IFN-γ, IL-1, IL-6, IL-10, TNF, CCL5, CXCL10, CXL11, and host factors such as VEGF [23,24,25,26,27]. The innate immune response is followed by an influx of cytotoxic T lymphocytes (CTLs) to the sites of infection that correlate with disease severity [14,28,29]. Natural killer (NK) cells are highly activated during infection, and their cell numbers in sera remain high for a long period (60 days) after the onset of disease symptoms [30,31]. Strong humoral responses are activated following infection as high viral protein-specific IgM and IgG antibodies titers are observed during the acute phase of infection [32,33,34,35].

The mechanisms used by hantaviruses to modulate the innate response in humans has been an active area of study. To probe virus–host interactions, surrogate cell types have been employed including A549 lung epithelial cells [36,37], HEK 293 cells [38,39,40], primary human umbilical cord vein endothelial cells (HUVECs) [39,41,42,43], and primary human LMVEC (HLMVEC) [44,45]. Together, the research suggests the hantaviral glycoproteins (GnGc) and the nucleocapsid (N) protein play key roles in the downregulation of the antiviral immune response. The Gn-tail of the glycoproteins (GnGc) and the nucleocapsid (N) protein have been implicated in inhibition of Toll-like receptors (TLRs) and retinoic acid-inducible gene I (RIG-I) cascades at multiple steps such as blocking TBK1 complex formation [39,46], NF-κB [47], and IRF nuclear translocation [39,40,46,47]. The inhibition of pathogen recognition receptor (PRR) signaling by the virus is essential for replication as hantaviruses are sensitive to IFN responses [39]. In contrast, pathogenic viruses such as Andes virus (*Orthohantavirus andesense*, ANDV), Hantaan virus (*Orthohantavirus hantanense*, HTNV), and Sin Nombre virus (*Orthohantavirus sinnombreense*, SNV) have been shown to delay the proinflammatory host response by blocking PRR signaling [38,39,41,44,46].

As stated above, prior studies have contributed greatly to our understanding of hantaviral–host interactions; however, a gap remains in the lack of measurements over time that would permit mathematical modeling of data. To further advance insight into the dynamics of the proinflammatory and anti-inflammatory responses of primary HLMVEC following infection with pathogenic and nonpathogenic hantaviruses, we established an approach for the study of infection and host response with ANDV, HTNV, and Prospect Hill virus (*Orthohantavirus prospectense*, PHV) over multiple time points. Initially, we established a primary HLMVEC donor (male) that had a similar level of infection for all three viruses to strengthen our comparisons. We evaluated 39 genes at six timepoints, although only a few were upregulated: *CCL5*, *CXCL10*, *CXCL11*, *IDO1*, *IRF7*, *TLR3*, and *IFNB1*. We confirmed secreted proteins levels for CCL5, CXCL10, CXCL11, IDO, and IFN-β. To further validate our findings, we expanded our studies to one additional male and two female donors. Notably, female donors produced higher levels of CXCL10, IDO, and IFN-β than males and moreover, the magnitude of the response was not similar across ANDV, HTNV, and PHV. Statistical pairwise analysis of protein levels suggests that virus species and individual HLMVEC characteristics, together, contribute to the outcomes. Lastly, we show that ANDV is capable of suppressing immune response involved with cell death in HLMVECs.

## 2. Materials and Methods

### 2.1. Cells and Viruses

HLMVECs (Cell Applications Inc., San Diego, CA, USA, or Promo Cell, Heidelberg, Germany) were cultured in HMVEC Growth Medium (Table 1). All experiments were performed with cells passaged four to six times. Primary HLMVECs were seeded at a density of 40,000 cells/cm^2^ on plates coated with collagen as specified by the manufacturer. All experiments were performed 12–16 h after seeding. ANDV (strain Chile-9717869), HTNV (strain 76–118), and PHV were a kind gift of Connie Schmaljohn at the United States Medical Research Institute of Infectious Diseases. Viruses were amplified in Vero E6 cells (ATCC CRL-1586) in Earls’ Modified Essential Medium (EMEM, Corning, Corning, NY, USA) with 10% FBS (Gibco, Waltham, MA, USA), 5 mM L-glutamine (Corning), and 1% penicillin–streptomycin (Gibco). Virus titers were determined by plaque assay [48]. Cells and viruses were checked for *Mycoplasma* spp. using LookOut Mycoplasma PCR Detection Kit (Sigma-Aldrich, St. Louis, MO, USA). All other reagents used were purchased from Thermo Fisher unless specified. All work with viruses was conducted at the UTHSC Regional Biocontainment Laboratory (RBL) in BSL-3.

### 2.2. Immunofluorescence Microscopy

HLMVECs, donor 2559, were seeded at a density of 40,000 cells/cm^2^ on glass coverslips in a 24-well plate and infected with ANDV, HTNV, or PHV at an MOI of 2 for 1 h and rocked every 15 min; after infection, cells were washed twice with Hank’s balanced saline solution (HBSS, Cell Applications Inc.) to remove residual virus from each well and incubated at 37 °C, 5% CO_2_. At 24 and 72 hpi, cells were washed with HBSS and fixed for 15 min at RT with 4% paraformaldehyde and 0.25 M sucrose. The paraformaldehyde solution was neutralized by washing the cells three times with 50 nM ammonium chloride. Cells were permeabilized with a 0.2% Triton-X, 1% bovine serum albumin (BSA) solution and blocked for 1 h with 5% goat calf serum, 0.2% Saponin. Cells were incubated with primary antibody in 0.2% Saponin, 1% BSA for 2 h at RT or overnight at 4 °C. For ANDV and HTNV, a rabbit polyclonal antibody against their N protein was used, and for PHV, a mouse monoclonal antibody (10R-2502, Fitzgerald, Acton, MA, USA) PHV N protein was used. Cells were incubated with anti-secondary antibody conjugated AlexaFlour 488 (Invitrogen, Carlsbad, CA, USA) for 1 h at RT. Cell nuclei were strained with 4′,6-diamidino-2-phenylindole (DAPI, Invitrogen) for 2 min. Coverslips were mounted, visualized, and cells with N protein were counted using a Zeiss 710 at the UTHSC Imaging Center and analyzed using Zeiss Zen (Black edition, version 2.3).

### 2.3. Gene Expression Assay

A custom QuantiGene Plex Assay (Invitrogen) consisting of 41 analytes containing two endogenous controls was used to measure gene transcript levels. The genes included in the panel were antiviral responses, *IRF3*, *IRF7*, *IFNA1*, *IFNB1*, *IFNG*, *NFKB1*, *TLR3*, *TLR7*; proinflammatory cytokines, *IL1A*, *IL1B*, *IL6*, *IL8*, *IL12A*, *IL15*, *TNF*; proinflammatory chemokines, *CCL2*, *CCL5 CCR7*, *CXCL10*, *CXCL11*; proinflammatory costimulatory molecules, *CD11B*, *CD14*, *CD200R1*, *CD80*; proinflammatory metabolites *ARG1* and NOS2; anti-inflammatory cytokines *IL1RN*, *IL10 TGFB1*; anti-inflammatory chemokine *CCL22*; anti-inflammatory costimulatory molecule, CD274; anti-inflammatory metabolites *IDO1* and *PPARG*; apoptosis and endothelial cell signaling; *CASP3*, *CASP7*, *CASP8*, *VEGFA*, and the endogenous controls *HPRT1* and *TBP*. HLMVECs, donor 2559, were seeded in 48-well plates precoated with collagen (Cell Application Inc.). HLMVECs, donor 2559, were infected with ANDV, HTNV, or PHV at an MOI of 2 for 1 h, rocked every 15 min, washed twice with HBSS to remove residual virus from each well, and incubated at 37 °C, 5% CO_2_. The virus and mock-infected cell monolayer were collected at 12, 24, 36, 48, 60, and 72 hpi using lysis mixture solutions (Invitrogen) and stored at −80 °C until completion of the assay, which was performed following the manufacturer’s instructions. Assay endpoints were measured on a MagPix (Luminex, Austin, TX, USA). Raw results were analyzed by subtracting values measured from water-substituted samples from all experimental sample values to which a constant value was added. All samples were normalized to endogenous controls and the mean from the biological repeats were calculated, which was used to calculate fold change and log_2_ transformed.

### 2.4. Multiplex Immunoassay

A custom ProcartaPlex immunoassay (Invitrogen) consisting of CCL5, CXCL10, CXCL11, IDO, and IFN-β five analytes were purchased from Invitrogen. HLMVECs, from four donors (Table 1), were seeded in collagen (Cell Application Inc.)-coated 48-well plates and infected with ANDV, HTNV, or PHV at an MOI of 2 for 1 h and rocked every 15 min; after infection, cells were washed twice with HBSS to remove residual virus and incubated at 37 °C, 5% CO_2_. At 48 and 60 hpi, supernatant was removed from wells and centrifuged for 5 min at 220× *g* to which Halt Protease Inhibitor Cocktail was added, and the assay was performed immediately following collection following the manufacturer’s instructions. The assay was read on a MAGPIX instrument (Luminex).

### 2.5. Caspase-3/7 Glo Activity Assay

HLMVECs, donor 2559, were seeded in collagen-coated 96-well plates and infected with ANDV at an MOI of 2 for 1 h and rocked every 15 min; after initial infection, cells were washed with HBSS to remove residual virus from each well and incubated at 37 °C, 5% CO_2_. At 60 h post infection, 1 µM staurosporine (Sigma-Aldrich) was added and incubated for 6 h. After incubation, Caspase-3/7 Glo was added to each well and incubated at RT for 30 min from which luminescence was read on an EnVision 2104 Multilabel Reader (PerkinElmer, Waltham, MA, USA).

### 2.6. Least Squares Fit

The results of the gene expression profiles of *CCL5*, *CXCL10*, *CXCL11*, *IDO*, and *IFNB1* measured in infected HLMVECs, donor 2559, by ANDV, HTNV, or PHV at 0, 12, 24, 36, 48, 60, and 72 hpi were fit to a mathematical model. The model predicts the time course of changes in gene expression profiles over 72 hpi. The fitted curves to the time series data of five gene profiles for ANDV, HTNV, and HTNV are discussed in Section 3.4. Appendix A is a summary of the results of fitting nonlinear least-squares regressions to the model. Each of the 15 fits (5 gene profiles and 3 viruses) based on the 7 observations at the time points *t* = 0, 12, 24, 36, 48, 60, and 72 hpi yield a minimum sum of squared error (SSE). The model fit is used to predict the time and the peak of the gene fold change. The margins of error for the model parameters pertaining to a 95% confidence interval and the R^2^ values calculated from the SSE provide a comparative assessment of the quality of the fits to the data. More details about the least squares fit are in Appendix A.

## 3. Results

### 3.1. Infection of a Male HLMVEC Donor by ANDV, HTNV, or PHV

The primary site of replication in human cases of HPS and HFRS are HLMVECs. We first asked if primary HLMVECs from a male, Caucasian 42-year-old donor, donor 2559, would support replication of ANDV, PHV, and HTNV equivocally to enable downstream comparisons. Cells were infected at an MOI of 2, and at 24 and 72 hpi cells were immunostained for N protein of each virus. The total number of cells was counted at 10× magnification along with the number cells that were infected at 24 and 72 hpi (Figure 1). At 24 hpi, cells had a similar number of infected cells, with ANDV having infected 50% of the cells whereas both HTNV and PHV infected 45% of the cells. A similar percentage of cells were infected at 72 hpi with no change seen from 24 hpi with ANDV having infected 46% of the cells, HTNV 44%, and PHV 39%. *T*-test analysis of the percentage of cells infected showed there were no significant differences in the percent of cells infected at 24 and 72 hpi for ANDV (*p* = 0.22), HTNV (*p* = 0.72), and PHV (*p* = 0.68). These results suggested that the donor would be appropriate for the proposed comparative studies.

### 3.2. Gene and Protein Expression Profiles from Male HLMVEC Donor following ANDV, HTNV, or PHV

To further advance mathematical analyses of the proinflammatory and anti-inflammatory responses of primary HLMVEC following infection with pathogenic and nonpathogenic hantaviruses, we evaluated the expression levels of 39 genes at 12, 24, 36, 48, 60, and 72 hpi (Figure 2, Appendix A). We infected HLMVECs, donor 2559, with ANDV, HTNV, or PHV at an MOI of 2. Expression of *CXCL11*, *IDO1*, *IFNB1*, and *TLR3* were greatest at 48 hpi for all three viruses, whereas the levels of *CCL5* peaked at 60 hpi (Figure 2). In ANDV-infected cells, *CXCL10* levels were highest at 36 hpi, which was 12 h prior to the peak levels of expression for HTNV- and PHV-infected cells. Peak levels of expression of *IRF7* differed for each virus; 36 hpi for ANDV, 60 hpi for PHV, and 72 hpi for HTNV. No increases in RNA levels were observed for *IFNA1*, *IFNG*, *IRF3*, *NFKB1*, *TLR7*, *TRL4*, *IL1A*, *IL1B*, *IL6*, *IL8*, *IL12A*, *IL15*, *TNF*, *CCR5*, *CD14*, *CD200R1*, *CD80*, *CD86*, *ITGAM*, *ARG1*, *NO2*, *IL1RA*, *IL10*, *TGFB1*, *CCL22*, *CD274, PPARG*, *CASP3*, *CASP7*, and *VEGFA*.

We used an ANOVA to test for differences in levels of *TLR3*, *IRF7*, *IFNB1*, *CCL5*, *CXCL10*, *CXCL11*, and *IDO1* upregulated by each virus for the six timepoints. *TLR3* reached peak levels of mRNA expression at 48 hpi following infection by each virus, and the mRNA levels were not statistically different to each other. At 60 hpi, mRNA levels induced by HTNV infection dropped but were significantly higher (*p* = 0.004) than those of ANDV. However, there was no significant difference at 60 hpi between HTNV and PHV. *TLR3* levels continued to drop for all three viruses at 72 hpi and were significantly higher for HTNV-infected cells in contrast to ANDV- and PHV-infected cells (*p* ≤ 0.0001 and 0.0145; respectively).

As expected, since *IFR7* is upregulated via *TLR3*, mRNA levels of *IRF7* were significantly different between HTNV and PHV (*p* = 0.036) at 48 hpi, and HTNV as compared to ANDV or PHV at 60 (*p* ≤ 0.0001 and 0.0004; respectively) and 72 (*p* ≤ 0.0001 and <0.0001; respectively) hpi. Intriguingly, *IRF7* transcript levels were significantly higher at 36 hpi in PHV-infected cells as compared to ANDV- and HTNV-infected cells (*p* = 0.0392 and 0.0001; respectively).

Levels of *CCL5* induced in HLMVECs were unique for each virus. ANDV was at 36 hpi, PHV at 60 hpi, and HTNV at 72 hpi. Levels of *CCL5* in ANDV-infected cells were significantly higher at 48 hpi compared to that of PHV-infected cells (*p* = 0.003), whereas at 72 hpi *CCL5* was greater in HTNV-infected cells as compared to ANDV- and PHV-infected cells (*p* = 0.0002 and 0.0004; respectively).

The mRNA levels of *CXCL10* peaked in ANDV-infected cells at 36 hpi and at 48 hpi in HTNV- and PHV-infected cells. At 36 hpi, levels of *CXCL10* were significantly lower in ANDV-infected cells as compared to HTNV- and PHV-infected cells (*p* = 0.0077 and 0.0062; respectively).

*CXCL11* levels reached peak mRNA expression at 48 hpi in all virus-infected cells, with no significant differences. *CXCL11* levels were decreased and were significantly lower in ANDV-infected cells as compared to HTNV- and PHV-infected cells at 60 (*p* = 0.0088 and 0.0394; respectively) and 72 hpi (*p* ≤ 0.0001 and 0.0063).

*IDO1* levels in ANDV-, HTNV-, and PHV-infected cells peaked at 48 hpi. However, PHV-infected cells’ levels of *IDO1* were significantly lower as compared to ANDV- and HTNV-infected cells (*p* = 0.0113 and 0.0107, respectively). Interestingly, *IDO1* levels in PHV-infected cells were significantly lower as compared to the levels of *IDO1* in ANDV-infected cells at 36 hpi (*p* = 0.0116).

Levels of *IFNB1* mRNA in ANDV-, HTNV-, and PHV-infected cells reached their peak at 48 hpi. ANDV-infected cells had significantly higher mRNA levels of *IFNB1* as compared to HTNV- and PHV-infected cells at both 36 hpi (*p* = 0.0004 and 0.0066, respectively) and 48 hpi (*p* = 0.0012 and 0.0161, respectively).

Following the evaluation of the gene expression profiles, we set to expand upon on it by measuring secreted protein levels for the genes that were observed to be upregulated above. HLMVECs, donor 2559, was infected with ANDV, HTNV, or PHV and the secreted protein levels of CCL5, CXCL10, CXCL11, IDO, and IFN-β measured at 48 and 60 hpi (Figure 3).

At 48 hpi, secreted protein levels of CCL5 revealed that PHV-infected cells secreted the same levels as ANDV- and HTNV-infected cells (*p* = 0.0526 and 0.99, respectively). However, protein levels differed between ANDV and HTNV (*p* = 0.04). Levels of IDO differed between PHV-infected cells and that of ANDV and HTNV-infected cells (*p* = 0.0032 and 0.081; respectively). The levels were similar between ANDV- and HTNV-infected cells. Infection by all three viruses resulted in similar levels of secreted protein levels of CXCL10, CXCL11, and IFN-β. Measurement of the secreted protein levels at 60 hpi revealed that each of the proteins, CCL5, CXCL10, CXCL11, IDO, and IFN-β, had similar secreted protein levels regardless of the virus.

### 3.3. Mathematical Model of the Gene Expression Profiles and Secreted Protein Levels

A mathematical model, a system of ordinary differential equations (ODEs), for gene expression within infected HLMVECs was formulated. In this model, we assume that bystander cells, specifically uninfected endothelial cells, are responsible for gene activation and that they are upregulated through cytokine signals from actively infected endothelial cells [49]. The goal is to provide a simple framework that allows comparison of the dynamics of the gene expression levels (fold-change) with respect to three different viral infections (ANDV, HTNV, and PHV). We do not model gene expression or protein production via a complex gene regulatory network, e.g., [50,51]. The underlying model hypothesis is that bystander cells play a significant role in the innate response as indicated by the gene expression data in the span of 72 h following viral infection. Data on gene-fold change for *IFNB1*, *CCL5*, *CXCL11*, CXCL10, and *IDO1* after experimental infection of HLMVEC by pathogenic (ANDV, HTNV) and nonpathogenic (PHV) viruses were recorded at seven timepoints *t* = 0, 12, 24, 36, 48, 60, 72 hpi. The data were fit via a nonlinear least squares procedure to the system of ODEs. To ensure that protein levels are also upregulated, additional data on protein levels associated with these particular genes are recorded at two time points *t* = 48, 60 hpi. A target-cell limited model serves as the framework for describing the infection dynamics at the cell level and drives the within-cell gene activation [52,53]. The target cells are the HLMVECs. Viral reproduction is not included in the model. Several assumptions are made based on the laboratory experiment and the model hypothesis regarding bystander activation: (1) total number of endothelial cells is constant, (2) cellular division does not occur, (3) actively infected cells trigger the transition of uninfected endothelial cells into bystander cells, (4) there is no gene interaction, and (5) the same model framework applies to all viruses. With these assumptions, the target-cell limited model is a system of ODEs with variables for *E* = number of uninfected endothelial cells, *I* = infected endothelial cells not producing virus, *I_A_* = infected endothelial cells actively producing virus, *B_i_* = uninfected bystander endothelial cells stage *i* that upregulate cytokine/chemokine signaling (dot over each variable represents the derivative with respect to time):(1)$$\dot{E}=-\rho I_{A}E\;\dot{I}=-f\left(t\right)I\;{\dot{I}}_{A}=f\left(t\right)I-\delta_{1}I_{A}(1)\;{\dot{B}}_{1}=\rho I_{A}E-\delta_{2}B_{1}\;{\dot{B}}_{i+1}=\delta_{2}\left(B_{i}-B_{i+1}\right),\;i=1,2,3.$$

Model (1) is based on the preceding assumptions and the experimental results. At the initiation of the experiment, the HLMVECs are either infected I or uninfected E. It is assumed that the infected target cells become activated and transition to cell stage IA. These cells signal neighboring uninfected cells through direct contact or paracrine signaling [49]. These neighboring uninfected cells are the bystander cells. Uninfected endothelial cells express receptors for interferons and for a wide range of cytokines and chemokines [54]. Model variables and parameters are defined in Table 2 and the biological meaning of the parameters in Table 3.

Parameter values were assigned for the target-cell model (1) through discussions among the modelers and experimentalists and from some preliminary model fits. An approximate delay of 24 h for within-host viral processing in stage *I* is assumed before actively infected cells *I_A_* begin signaling bystander cells. In particular, the per capita rate of transfer to active infection is a logistic curve with *t* = 24 the time at which the transfer reaches half-maximum rate, *f*(24) = 0.5*γ*, with maximal rate *γ* = 1/6:(2)$$f\left(t\right)=\frac{\gamma }{1+exp(-0.5\left(t-24)\right)}$$

The actively infected cells *I_A_* trigger production of bystander cells for an average duration of 12 h. To distinguish early from late bystander upregulation of genes, the duration of bystander cell signaling was divided into four stages, each of average duration 4 h, *B*_1_ → *B*_2_ → *B*_3_ → *B*_4_, 1/δ_2_ = 4 h. More details about the dynamics of model (1) are discussed in the Appendix A.

The data were fit to the five genes studied in this investigation. Variables G_1_ through G_5_ denote the genes in the respective order, *IFNB1*, *CCL5*, *CXCL11*, *CXCL10*, and *IDO1*. The fold-change in *CCL5* occurred much later than the other genes. Therefore, it was assumed that *CCL5* was upregulated during bystander stages *B*_3_ and *B*_4_, while the remaining genes were upregulated during stages *B*_1_ and *B*_2_. For gene data corresponding to each virus, ANDV, HTNV, and PHV, the least squares fit (Figure 4) was used to estimate three parameters, two representing gene upregulation or production rate: *a_i_*_1_ and *a_i_*_2_ from two bystander stages and a third representing the degradation rate, *r_i_*, *i* = 1, 2, 3, 4, 5. The target-cell limited model (1) is coupled with the gene model (3):(3)$${\dot{G}}_{i}=a_{i1}B_{1}+a_{i2}B_{2}-r_{i}G_{i},\;\;i=1,3,4,5\;{\dot{G}}_{2}=a_{21}B_{3}+a_{22}B_{4}-r_{2}G_{2}.$$

The dynamics of the target-cell limited model (1) and their relation to the gene model (3) are illustrated by a compartmental diagram (Appendix A).

The three fitted curves for ANDV, HTNV, and PHV are plotted together (Appendix A) to show the trends in gene expression profiles that are associated with the virus. The fitted curves show the same trends as the data plotted in Figure 2.

Appendix A summarizes the SSEs for the model fits, the times and the values of the peaks for each of the fitted curves, the margins of error pertaining to a 95% confidence interval for each parameter, and R^2^ values computed from the SSE. The margins of error are calculated based on the asymptotic distribution of the parameter estimates [55].

These are seemingly vastly different fits, with SSE values ranging over five orders of magnitude. The reason for this is the corresponding widely ranging fold changes. If one computes the R^2^ values, which in linear regression provide an informative measure of the fitted curve by scaling the SSE by the variance of the data^2^, we obtain values in the tight range of 0.95 to 1.00. Although R^2^ is not appropriate in nonlinear regression for assessing the proportion of variability explained by the fit, it is useful for providing a comparative assessment of the quality of the fits. These high R^2^ values are indicative of some degree of overfitting, as evidenced by the close adherence to the observed values. However, the large number of parameters is needed in the system of ODEs to adequately capture the nature of the fold change dynamics.

### 3.4. Immune Responses of Male and Female Donors following Infection

We hypothesized that donor characteristics may potentially confound host response to ANDV, HTNV, and PHV. Hence, we obtained two female and one additional male donor for HLMVEC, resulting in two donors from males ages 19 and 42 and two donors from female ages 34 and 53 (Table 1). Based on the prior results reported above, we measured protein levels of CCL5, CXCL10, CXCL11, IDO, and IFN-β at 48 and 60 hpi. Each donor was infected with ANDV, HTNV, or PHV at an MOI of 2.

Secreted protein levels measured during infection of each of the four HLMVEC donors by ANDV, HTNV, or PHV showed that each of the five measured proteins were upregulated at 48 and 60 hpi (Appendix A). ANOVA of these data suggested the donor characteristic and virus species contribute to the heterogenous immune responses observed during infection of the four donors during infection by each hantavirus species. Stratifying measured protein levels by donor sexes and analyzing secreted protein levels by ANOVA revealed in female donor cells, specific virus species induced higher levels of CXCL10 (ANDV at 48 hpi), IDO (ANDV at 48–60 hpi; PHV at 60 hpi), and IFN-β (ANDV, HTNV at 60 hpi) (Figure 5). Statistical analysis showed that HLMVEC donor characteristics as well as virus species together contributed to differences observed in secreted protein levels (Appendix A).

### 3.5. ANDV Infection Inhibits Cell Death

Hantavirus infection of cells does not cause any cytopathic effect; with this, previous studies on apoptosis during hantaviral infection in Vero E6 cells and HUVECs revealed that hantaviruses are able to inhibit cell death of infected cells through the suppression of immune responses and inhibition of caspase-driven apoptosis. Based on these findings, we expected hantavirus-infected HLMVECs to inhibit apoptosis following induction of cell death by staurosporine. To examine whether hantaviruses inhibit apoptosis in infected HLMVECs, HLMVECs from donor 2559 were infected with ANDV at an MOI of 2.

At 60 hpi, cells were treated with staurosporine (ST) to induce apoptosis. At 6 h post-ST addition, caspase 3/7 activity was measured (Figure 6). HLMVEC treatment with ST was able to induce apoptosis of cells as a five-fold increase of caspase activity was observed. ST treatment of ANDV-infected cells caused approximately a two-fold increase when compared to that of ANDV-infected HLMVECs. When comparing ANDV- and mock-infected HLMVECs following ST treatment, there was a two-fold reduction in caspase activity.

## 4. Discussion

In this study, we determine that three hantavirus species ANDV, HTNV, and PHV that drive different disease outcomes in humans each induce unique immune responses during infection of primary HLMVECs. It has long been hypothesized that the differences in the activation of immune responses between hantaviruses are responsible for the disease outcomes that are observed in humans. These differences have lent themselves to be an active area of hantavirus research, but study findings can be difficult to translate or to draw concrete conclusions, which may be caused by several factors used in study models such as the hantavirus species and surrogate cell types. To better understand the early antiviral and innate immune responses induced in HLMVECs during infection by ANDV, HTNV, or PHV and how they differ between hantavirus species, we defined these responses through measurement of host gene expression profiles and secreted protein as well as mathematical models.

First, we determined that HLMVECs from one donor can be infected by each of the three virus species by immunofluorescent microscopy and staining viral N protein. This revealed that ANDV, HTNV, and PHV infected a similar number of cells at 24 and 72 hpi. These findings show that the HLMVEC donor we selected was suitable to study immune responses. Similar number of cells infected between virus species suggests that differences in immune responses observed were likely not driven by differences in the number of HLMVECs infected.

Antiviral and immune responses were first measured at a gene expression level and the genes, *TLR3*, *IRF7*, *IFNB1*, *CCL5*, *IDO1*, *CXCL10*, and *CXCL11*, were upregulated with similar pattern between virus species. These genes have previously been reported to be upregulated during hantaviral infection studies using in vitro and in vivo models. Of these genes, the RNA sensing receptor, *TLR3*, is induced during HTNV infection but not by that of PHV in Huh7 cells [37]. The expression of the receptor has also been reported in primary cell HUVECs, but expression differed based on the measurement method used; induction could only be measured by qRT-PCR and not microarray [43]. During in vivo infection of Syrian golden hamsters by ANDV and HTNV, both trigger TRL3 induction [56]. Downstream within the TLR-signaling pathways are the IRF proteins, and of these proteins, specifically IRF3 is shown to be of great importance during hantavirus infection [41,43,44]. In contrast to previous studies, we did not observe *IRF3* induction but rather that of *IRF7*. IRF7 induction has been noted in SNV infection of HLMVECs and during infection of Syrian golden hamsters where there was a similar lack of IRF3 induction with upregulation of *IRF7* [45,56]. With the induction of *IRF7*, it is expected that IFN-β expression follows, and previous studies on HLMVEC determine PHV significantly upregulates the gene and protein expression when compared to that of ANDV [44]. Our modeling of *IFNB1* suggests contradictory findings in that ANDV infection induced higher levels of *IFNB1* gene expression with peak levels being reached earlier as compared to PHV.

The three chemokines, CCL5, CXCL10, and CXCL11, that were upregulated may explain the infiltration and activity of various immune cells such as macrophages, monocytes, and dendritic cells during human hantaviral infection. Hantavirus species comparative studies performed in HUVECs show similar gene expression profiles for *CCL5* during HTNV and PHV infection, whereas both *CXCL10* and *CXCL11* transcript levels are expressed the highest following infection of HTNV followed by PHV [41]. Our modeling of these three genes indicates that HTNV infection resulted in the strongest activation of *CCL5* and *CXCL11* compared to ANDV and PHV, whereas ANDV infection caused the strongest *CXCL11* activation. These chemokines are defined in HPS patients’ sera and exhibit different trends in that CCL5 is downregulated compared to control groups and decreases with disease progression. CXCL10 on the other hand is upregulated and increases with the stage progression, whereas no significant differences in CXCL11 were found between HPS and control patients [25].

The expression of IDO1 and its secretion during hantavirus infection is novel as it has only been previously reported during human cases of nephropathia epidemica (NE), a milder form of HFRS caused by *Puumala orthohantavirus* infections [57,58]. In serum analysis of NE patients, elevated levels of IDO were reported, and these levels increased as disease phases progressed. High IDO levels also correlate with disease severity [57]. Relationship modeling revealed that high IDO levels correlate with disease severity and that the protein’s activity correlates with FOXP3 expression in regulatory T cells leading to immune suppression and potentially preventing viral clearance [58]. IDO is mostly secreted by immune cells such as macrophages, monocytes, and dendritic cells in response to IFN-γ. Along with immune cells, IDO can be expressed by endothelial cells. While IDO is largely a protein of focus in cancer biology, its activity has been noted for a number of RNA viruses such as Dengue virus, hepatitis-C virus, and influenza A, where increased level is observed following infection by the viruses. The expression and secretion of IDO could potentially play a role in the lack of viral clearance function of immune cells during infection of hantaviruses [59,60,61,62].

Of interest are the genes that were not upregulated in gene expression analysis; genes such as *IL6* and *VEGFA*. IL-6 is viewed as a critical cytokine during hantaviral infection, and the proteins’ serum levels correlates with disease severity in human HPS cases. Unique induction of the gene is observed in HUVECs based on virus species as only HTNV but not PHV and SNV induces it a gene level; in another study with the same cell type, *IL6* expression is observed in HTNV- and PHV-infected cells but not ANDV [25,41]. Another critical cytokine, VEGF, is found at significantly higher levels in the pulmonary edema fluids of HPS patients, and these high protein levels correlate to disease severity [26].

Previously, Ontiveros et al. [47] reported that HTNV is capable of modulating host immune responses and inhibiting cell death induced by staurosporine in infected Vero E6 cells. The mechanism through which HTNV was able to inhibit cell death was shown to be through sequestering NF-κB in the cytoplasm and blocking effector caspases [47]. Hantaviral regulation of apoptosis was also studied in HUVECs and HLMVECs where it was found that a number of hantavirus species from the Old and New World are able to inhibit apoptosis through various mechanisms such as the down regulation of granzyme B activity, B-cell lymphoma-2 (BCL-2), and Death Receptor 5 (DR5/TRAIL receptor 2) [63,64]. In the current study, we showed that ANDV infection inhibited apoptosis in infected HLMVECs following treatment with staurosporine, showing this response is not cell-specific.

An important finding of this current study is the observation that hantavirus species drive unique responses in HLMVECs from different donors based on their characteristics. Human demographic studies of hantavirus-diseased patients have indicated that mortality rates are higher for females with HRFS in China and with HPS in Argentina [65,66]. It is of interest that stronger induction of CXCL10, IDO, and IFN-β was observed in female donors compared to males but also the finding that this induction was unique to particular virus species. The differences between secreted protein levels between HLMVEC donors could potentially be multifactorial as the donors differed on multiple aspects such as ages, sex, and race; as such, these findings highlight the need to study host responses in primary cell models consisting of multiple donors.

## Figures and Tables

**Figure 1 viruses-15-01806-f001:**
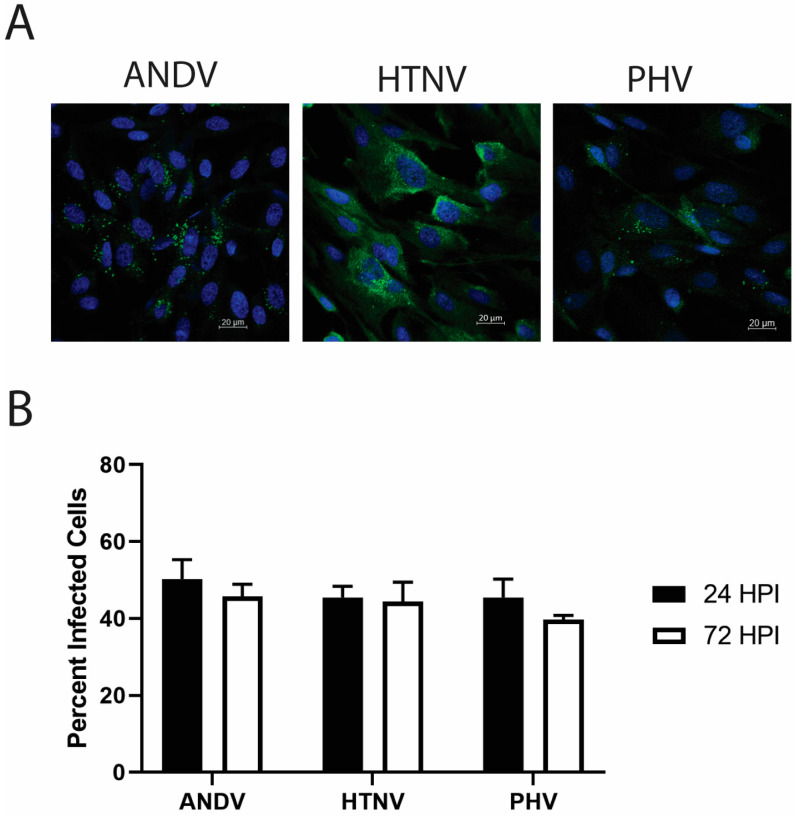
HLMVECs infected by ANDV, HTNV, or PHV at 24 and 72 hpi. HLMVECs, donor 2559, were infected with ANDV, HTNV, or PHV at an MOI of 2. At 24 and 72 hpi, infected HLMVECs were fixed and immunofluorescent stained for cells’ nuclei with DAPI and virus nucleocapsid protein antibody and visualized on a confocal microscope. (**A**) HLMVECs infected with ANDV, HTNV, or PHV at 72 hpi. (**B**) The total number of cells and the total number of infected cells were counted and the precent infected cells were calculated at 24 (black bars) and 72 hpi (white bars). Mean with standard deviation.

**Figure 2 viruses-15-01806-f002:**
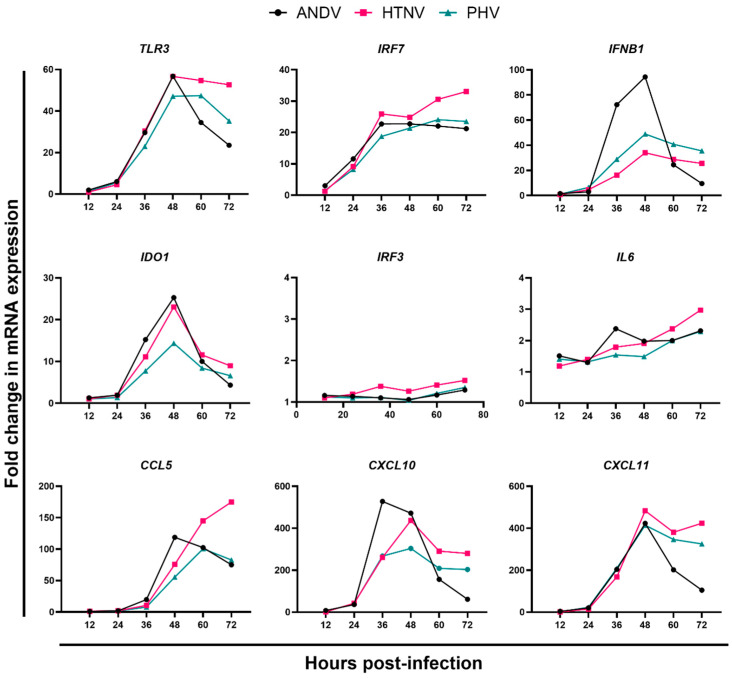
Selected PRR, cytokines, and chemokines showing upregulation at 12, 24, 36, 48, 60, and 72 hpi in HLMVECs following infection ANDV, HTNV, or PHV. HLMVECs, donor 2559, were mock-infected or infected with ANDV (black circles), HTNV (pink squares), or PHV (teal triangles) with an MOI of 2. At 12, 24, 36, 48, 60, and 72 hpi, cell lysates were collected and gene expression measured with a custom QuantiGene Plex Assay. To calculated gene expression, background fluorescent activity was subtracted from the fluorescent activity of each gene in each sample and a constant value was added. Fluorescent activity was log_2_ transformed and normalized to an endogenous control gene in each sample. Normalized values were used to calculate fold change in mRNA levels as compared to mock-infected samples.

**Figure 3 viruses-15-01806-f003:**
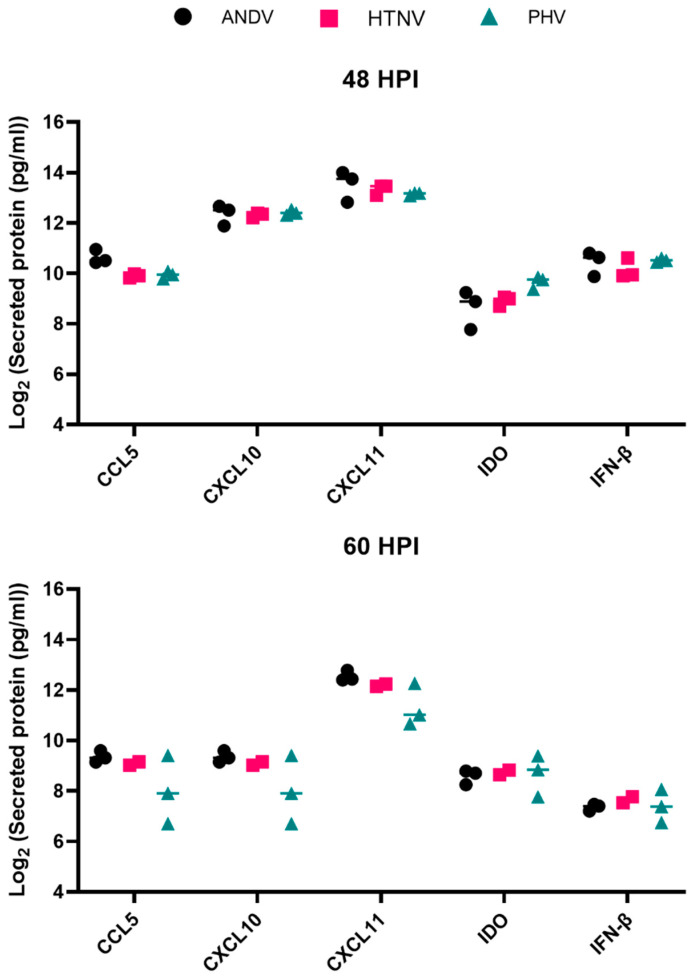
Select IFN, chemokine, and cytokine secretion protein levels at 48 and 60 hpi by HLMVECs following infection by ANDV, HTNV, and PHV. HLMVECs, donor 2559, were mock-infected or infected with ANDV (black circles), HTNV (pink squares), or PHV (teal triangles) at an MOI of 2. At 48 and 60 hpi, supernatants were collected and secreted protein levels measured. Measured values of each secreted protein were log_2_ transformed.

**Figure 4 viruses-15-01806-f004:**
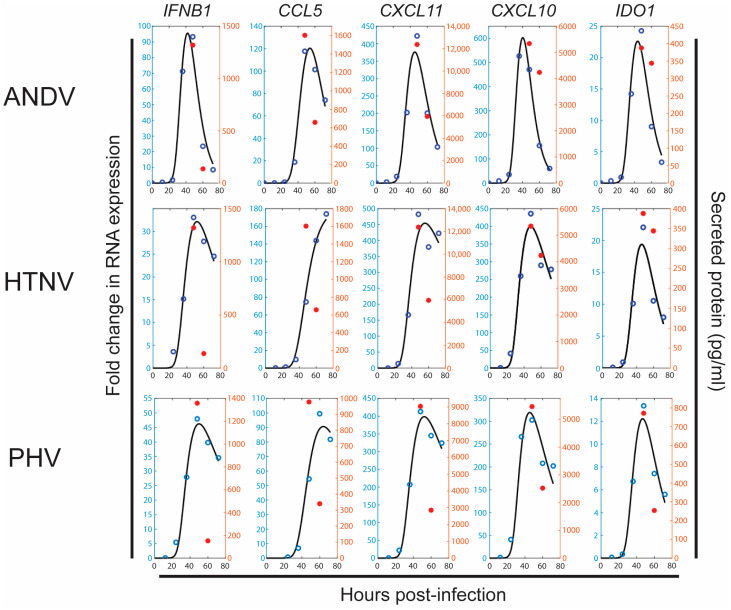
Models fit to data on cytokine and chemokine upregulation. Least squares fit to model (3) for *IFNB1*, *CCL5*, *CXCL10*, *CXCL11*, and *IDO1* at *t* = 0, 12, 24, 36, 48, 60, 72 hpi (data marked with blue circles and fitted curves in black). Two protein data points (pg/mL) at *t* = 48, 60 hpi are also plotted (marked with red asterisk).

**Figure 5 viruses-15-01806-f005:**
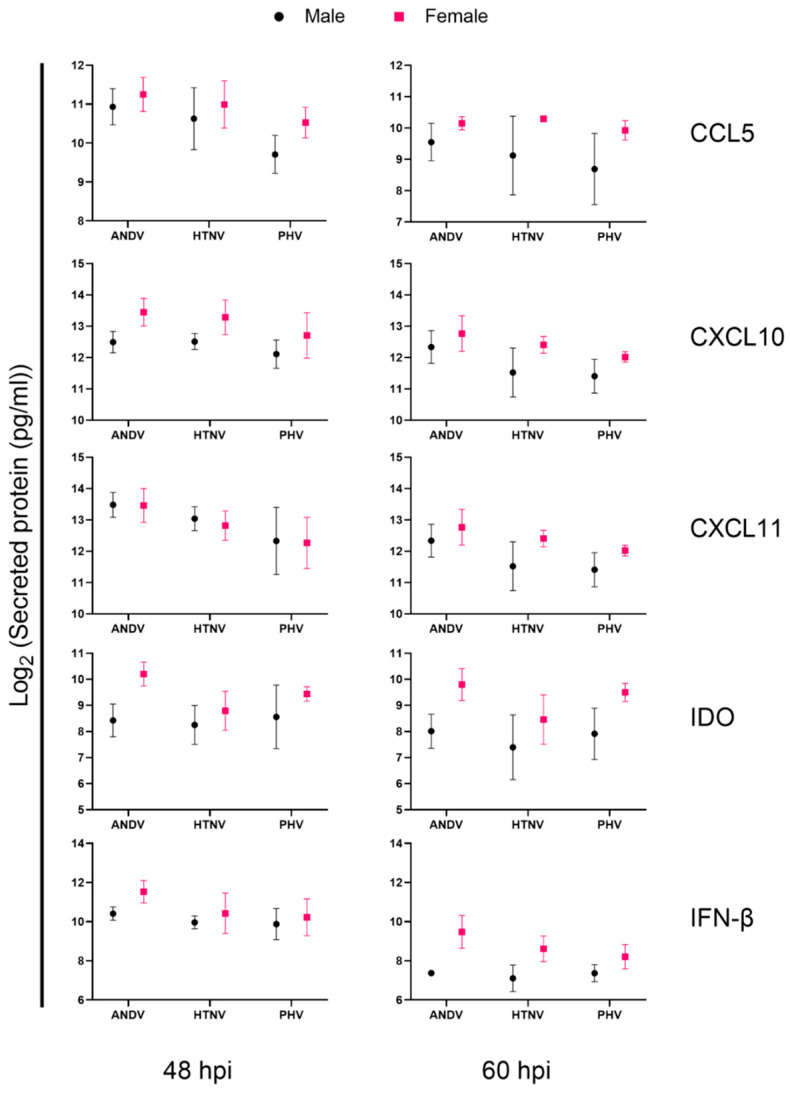
Select IFN, chemokine, and cytokine secretion protein levels at 48 and 60 hpi by male and female HLMVECs donors during infection by ANDV, HTNV, and PHV. HLMVECs from four donors, two males (black circles) and females (pink squares), were mock-infected or infected with ANDV, HTNV, or PHV at an MOI of 2. At 48 and 60 hpi, supernatants were collected and secreted protein levels were measured. Measured values of each secreted protein were log_2_ transformed and used to calculate fold change in secreted protein levels as compared to mock-infected samples. Fold change values measured from HLMVEC donors from the same sex were combined for each individual virus studied.

**Figure 6 viruses-15-01806-f006:**
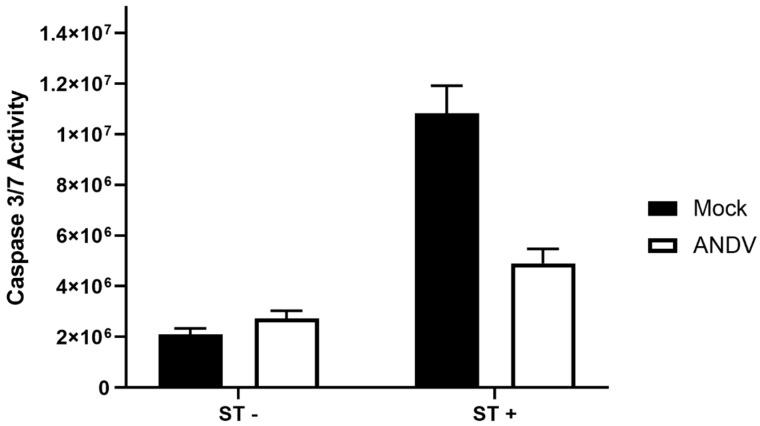
The inhibition of HLMVEC cell death by ANDV following staurosporine treatment. HLMVECs, donor 2559, were mock-infected (black bars) or infected with ANDV (white bars) at an MOI of 1.0. At 60 hpi, cells were treated with 1 µM staurosporine (ST) and after six hours post addition of ST, caspase 3/7 activity was measured by Caspase Glo 3/7 on a luminometer. Mean with standard deviation.

**Table 1 viruses-15-01806-t001:** Characteristics of HLMVEC donors.

	2559	2572	2551	438Z013.1
**Sex**	Male	Male	Female	Female
**Age**	42	19	34	53
**Race**	Caucasian	Hispanic	Hispanic	Caucasian
**Vendor**	Cell App Inc.	Cell App Inc.	Cell App Inc.	Promo Cell

**Table 2 viruses-15-01806-t002:** Variables and parameters for model (1) and (3).

Variable	Description	Parameter	Value
E	Uninfected endothelial cell	*ρ*	10^−5^
*I*	Infected endothelial cell	*γ*	1/6
*I_A_*	Actively infected endothelial cell	*δ* _1_	1/12
*B_i_*	Bystander endothelial cell stage *i*	*δ* _2_	1/4
*G_i_*	Activated gene *i*	*a_ij_*, *r_i_*	Estimated

**Table 3 viruses-15-01806-t003:** Parameter units and descriptions for models (1) and (3). Estimated and assumed parameter values are given in the text. hpi = hours post-infection; *I_A_* = actively infected endothelial cells; *B_i_* = uninfected bystander cells in stage *i* = 1, 2, 3, 4; *G_j_* = gene expression level within bystander cells, *j* = 1, 2, 3, 4, 5.

Parameter	Description	Units
*f*(*t*)	Time-dependent activation rate of infected cells *I*	hpi^−1^
*γ*	Maximum transition rate to actively infected cell	hpi^−1^
*ρ*	External signaling rate per actively infected cell *I_A_*	(hpi × infected cell) ^−1^
*δ* _1_	Rate external signaling decreases in actively infected cell *I_A_*	hpi^−1^
*δ* _2_	Transition rate between bystander cell stages	hpi^−1^
*a_ij_*, *j* = 1, 3, 4, 5 *i* = 1, 2	Fold change rate in gene *G_j_* within a bystander cell *B_i_*	(fold change)/(hpi × bystander cell)
*a_i_*_2_, *i* = 1, 2	Fold change rate in gene *G*_2_ within a bystander cell *B_i+_*_2_	(fold change)/(hpi × bystander cell)
*r_i_*, i = 1, 2, 3, 4, 5	Rate gene expression level *G_i_* decreased within bystander cells	hpi^−1^

## Data Availability

All data represented or discussed in this manuscript is available in Appendix A or upon request.

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
