# Peer review of "Modeling the Immune Response for Pathogenic and Nonpathogenic Orthohantavirus Infections in Human Lung Microvasculature Endothelial Cells"

_viruses, 2023, doi:10.3390/v15091806_

Round 1

Reviewer 1 Report

In this manuscript, the authors aimed to evaluate the temporal dynamics of host immune response modulation in primary human LMVECs following infection by Prospect Hill (nonpathogenic), Andes (pathogenic) and Hantaan (pathogenic) viruses. Overall there was no significant difference in the transcriptional profiles of key genes in primary HLMVECs following infection by pathogenic and nonpathogenic hantaviruses. Further experiments may be required and by doing so the quality of this manuscript will be improved greatly.

1)Figure 1: The authors need to show the representative images

2)It will be nice if the authors can figure out whether TLR3 pathway is responsible for IFN induction. For this, you will need to introduce knockdown or knockout TLR3 related genes in HLMVECs.

3)Although it is interesting to compare the immune response following infection with non pathogeneic vs. pathogenic viruses, the viral replication kinetics may differ. The authors need to show viral replication kinetics (using plaque assay) following the infection with three viruses

4)The rationale for Figure 6 is not clear. If the authors want to say that ANDV inhibits cell death, the authors need to do more compelling experiments regarding the type of cell death. Is it apoptosis or necroptosis? Also the authors need to show the cleaved form of caspases by immunoblotting.

Reviewer 2 Report

In the manuscript, the authors analyzed the modulation of host immune responses in human microvascular lung endothelial cells (LMVECs) after infection with the pathogenic orthohantaviruses Andes and Hantaan and the nonpathogenic Prospect Hill virus. The researchers measured RNA transcript levels of genes involved in antiviral, proinflammatory, anti-inflammatory, and metabolic signaling pathways at different time points. Gene expression analysis and mathematical modeling revealed similar profiles among the three viruses, with upregulated genes related to interferon signaling, immune cell recruitment, and immune response modulation. Protein secretion analysis revealed that the amount of certain secreted proteins varied between the viruses and the different donors. Overall, the study found that the transcriptional profiles of key genes in primary LMVECs were similar regardless of hantavirus pathogenicity, whereas protein secretion levels were more specific to individual donors than to virus type.

The manuscript is interesting and well written. The results of the study are important for a better understanding of hantavirus immunopathogenesis. I believe that the manuscript is suitable for publication in the journal Viruses. However, I have some minor comments and questions for the authors.

1. HLMVECs were infected with virus at a MOI of 2. What was the concentration of cells seeded onto a 24-well plate? Were they always seeded at a density of 40,000 cells/cm2?

2. Table S2 is well laid out and understandable to the reader. However, it would be easier for the reader if significant values of genes were in bold. I think it is easier for the reader if significant differences are not only in the text but also visualized in the table.

3. Are the values for "fold changes" in the graphs in Figure 2 single values or was the experiment performed in duplicate?

Reviewer 3 Report

This paper deals with modeling the immune response for pathogenic and nonpathogenic orthohantavirus infections in human lung microvascular endothelial cells (LMVEC). It can be accepted for the publication after some major revisions.

1. Recall the definition of RNA virus and its deference with DNA virus.

2. The construction of model (1) should be justified.

3. Each equation of models (1) and (3) should be more explained.

4. In model (1), check the expression of fourth equation. What means B_A?

5. Give the biological meanings of each parameter of models (1) and (3).

6. Study the dynamical behaviors of model (1).

7. Compare your results with others existing in the literature.

Check the language and punctuation of the manuscript.

Round 2

Reviewer 1 Report

The authors have addressed my concerns and the manuscript is greatly improved for acceptance. 

Reviewer 3 Report

The paper can be accepted.

The paper can be accepted.